# Cardiovascular Diseases in Public Health: Chromosomal Abnormalities in Congenital Heart Disease Causing Sudden Cardiac Death in Children

**DOI:** 10.3390/medicina60121976

**Published:** 2024-12-01

**Authors:** Cecilia Salzillo, Marco La Verde, Amalia Imparato, Rossella Molitierno, Stefano Lucà, Francesca Pagliuca, Andrea Marzullo

**Affiliations:** 1PhD Course in Public Health, Department of Experimental Medicine, University of Campania “Luigi Vanvitelli”, 80138 Naples, Italy; stefano.luca@unicampania.it; 2Pathology Unit, Department of Precision and Regenerative Medicine and Ionian Area, University of Bari “Aldo Moro”, 70121 Bari, Italy; 3Department of Woman, Child and General and Specialized Surgery, University of Campania “Luigi Vanvitelli”, 80138 Naples, Italy; marco.laverde@unicampania.it (M.L.V.); amalia.imparato@studenti.unicampania.it (A.I.); rossella.molitierno@studenti.unicampania.it (R.M.); 4Pathology Unit, Department of Mental and Physical Health and Preventive Medicine, University of Campania “Luigi Vanvitelli”, 80138 Naples, Italy; francesca.pagliuca@unicampania.it

**Keywords:** chromosomal abnormalities, genetic disorders, congenital heart disease, sudden cardiac death, cardiovascular diseases, public health

## Abstract

Chromosomal abnormalities (CAs) are changes in the number or structure of chromosomes, manifested as alterations in the total number of chromosomes or as structural abnormalities involving the loss, duplication, or rearrangement of chromosomal segments. CAs can be inherited or can occur spontaneously, leading to congenital malformations and genetic diseases. CAs associated with cardiovascular diseases cause structural or functional alterations of the heart, affecting the cardiac chambers, valves, coronary arteries, aorta, and cardiac conduction, thus increasing the likelihood of arrhythmias, cardiac arrest, and sudden cardiac death (SCD). An early diagnosis and the adequate management of chromosomal abnormalities associated with cardiovascular diseases are essential to prevent SCD, which is a serious public health problem today. In our review, we analyzed the structural and functional CAs responsible for congenital heart disease (CHD) that increase the risk of SCD and analyzed the prevention strategies to be implemented to reduce SCD.

## 1. Introduction

CAs are numerical or structural alterations of chromosomes. Humans have forty-six chromosomes divided into twenty-three pairs, twenty-two of which are autosomes (nonsex chromosomes) and one of which is a sex chromosome; these chromosomes vary according to sex: XX in females and XY in males.

CA can also cause cardiovascular disease (CVD) and is often associated with complex genetic syndromes that include cardiovascular manifestations such as CHD. These syndromes may also involve anomalies of facial structures and the musculoskeletal system and can lead to SCD.

CHDs represent a major cause of mortality and morbidity in children, significantly impacting public health globally, with a clinical impact ranging from mild cases to conditions requiring complex surgical interventions. Among the most serious complications associated with these malformations, SCD in children constitutes a problem of considerable importance, not only for the medical implications but also for the psychological and social burden that weighs on families and the healthcare system [1].

CHD occurs in 0.8 to 1.3% of live births [2] and is associated with extracardiac anomalies in 30% of cases [3]. The epidemiology of SCD in CHD is complicated by geographic variations, errors in death certificates, a lack of autopsies, and inaccuracies in diagnostic codes [4]. SCD is relatively rare in the general population, with an annual incidence between 0.07 and 0.40 per 100,000 person-years [5,6]. However, in CHD patients, the rates are 20–30 times higher than those in the general population [4]. Some patients with tetralogy of Fallot, transposition of the great arteries, cyanotic heart disease, Ebstein anomaly, and Fontan circulation are at an increased risk for SCD. The incidence of SCD is greater in adults than in children [7] and may be the highest in patients with complex CHD in their mid-30s [4,8].

An increasing number of newborns with CHD have excellent long-term survival due to advances in surgical procedures. However, the prognosis is strongly affected by the presence of extracardiac malformations and the underlying cause of the condition [9]. CHD associated with genetic abnormalities significantly affects parents’ decisions to opt for compassionate postnatal care or termination of pregnancy; consequently, genetic testing of fetuses with CHD is critical to facilitate both prenatal and postnatal management [1].

The importance of targeting CAs in research on CHD and SCD derives from the possibility of identifying predictive genetic biomarkers and developing personalized prevention strategies. Furthermore, understanding the genetic mechanisms underlying CHD could facilitate the development of targeted therapies and improve the early identification of high-risk patients, helping to reduce the health and economic burden of these diseases, and implement preventative strategies to reduce SCD risk.

This review article aims to analyze the literature on the structural and functional CAs responsible for cardiovascular diseases such as CHD, with a particular focus on SCD in children. Furthermore, we intend to discuss the clinical implications and outline future research directions, highlighting the importance of addressing these issues to improve public health and therapeutic prospects.

## 2. Classification

CAs are classified into numerical chromosomal abnormalities and structural chromosomal abnormalities.

Numerical chromosomal abnormalities are characterized by a variation in the number of chromosomes. These abnormalities are often associated with various cardiovascular diseases (Table 1) and include the following:

Aneuploidy: an anomalous number of chromosomes:

Trisomy: an extra chromosome, such as trisomy 21 or Down syndrome, trisomy 18 or Edwards syndrome, and trisomy 13 or Patau syndrome;

Monosomy: lack of a chromosome, such as monosomy X syndrome or Turner syndrome.

Structural chromosomal abnormalities are characterized by changes in the structure of one or more chromosomes. These abnormalities are often associated with various cardiovascular diseases (Table 1) and include the following:

Microdeletion: a small deletion that can be difficult to detect with standard techniques, such as DiGeorge syndrome characterized by 22q11.2 deletion;

Deletion: the loss of a portion of a chromosome, such as Williams syndrome characterized by 7q11.23 deletion.

These abnormalities can alter the normal development of the heart and blood vessels, leading to a variety of congenital heart defects that increase the risk of cardiovascular disease and, in some cases, SCD.

## 3. Materials and Methods

This review was conducted using PubMed and Scopus with the following search keywords: “Trisomy 21 OR Down syndrome”, “Trisomy 18 OR Edward syndrome”, “Trisomy 13 OR Patau syndrome”, “Monosomy X syndrome OR Turner syndrome”, “22q11.2 deletion OR DiGeorge syndrome”, “q11.23 deletion OR Williams syndrome”, “congenital heart disease AND sudden cardiac death AND children”, with inclusion criteria being primary studies, secondary studies, and English language. Additionally, Google Scholar was used for gray literature. ChatGPT4 was used for the translation of some paragraphs into English.

## 4. Numerical Chromosomal Abnormalities

### 4.1. Trisomy 21 or Down Syndrome

Down syndrome (DS), also known as trisomy 21 (Table 2), is a numerical chromosomal anomaly characterized by three chromosome 21 s and is the most frequent chromosomal anomaly, occurring in 16 cases per 10,000 live births [10].

SD is also the chromosomal anomaly most frequently associated with CHD, such as atrioventricular septal defects (AVSDs), ventricular septal defects (VSDs), atrial septal defects (ASDs), patent ductus arteriosus (PDA), and tetralogy of Fallot (TF), and with an increased incidence of coronary events such as myocardial infarction [11].

Compared with the general population, DS is associated with a 40–50 times greater probability of developing CHD [10,12]. In studies reported in the literature [13,14], AVSD and VSD account for 76% of CHDs in patients with DS. Approximately half of live-born infants with DS are diagnosed with CHD, compared with approximately 1% in the general population; however, the precise incidence of CHD is unclear, and the incidence reported in several studies in the literature [15,16,17,18] varies widely over time and in different places, ranging between 23% and 79%. In studies using diagnostic ultrasonography, CHD has been observed in 29–56% of cases of karyotype-proven DS [10]. AVSD is the most common, followed by isolated TF in 13%, AVSD and TF combined in 9%, and isolated VSD in 4–17% [10].

Additionally, several environmental factors increase the risk of CHD in DS, such as maternal smoking, obesity, and folic acid deficiency during pregnancy [18].

For prenatal diagnosis, ultrasound screening in the second trimester of pregnancy, between 18 and 22 weeks, is recommended. This test has an excellent CHD detection rate, reducing the need for fetal echocardiography [10].

Detailed fetal echocardiography is indicated if a fetal ultrasound suggests cardiovascular abnormalities, or, in the presence of conditions such as maternal diabetes, uncontrolled phenylketonuria, first-trimester rubella, fetal karyotype abnormalities, hydrops or fetal effusions, specific maternal medications, or a strong family history of CHD, it can accurately identify more than 90% of complex CHDs [19]. Moreover, a follow-up ultrasound and echocardiographic re-evaluation in the third trimester of pregnancy are performed to effectively manage the fetuses at risk and monitor evolving cardiac conditions [20,21].

Prenatal genetic testing consists of metaphase chromosome banding of fetal cells obtained through amniocentesis or chorionic villus sampling, but in recent years, fluorescence in situ hybridization (FISH) has become the preferred technique for detecting chromosomal abnormalities.

Currently, non-invasive prenatal tests, which use a combination of fetal ultrasound for nuchal translucency, maternal blood analysis, or cell-free DNA, were adopted for the first-trimester screening [22,23]. However, access and the rate of adoption for first-trimester screening varies widely with socioeconomic status and financial resources [24].

Amniocentesis and chorionic villus sampling (invasive tests) are reserved for high-risk patients.

The early identification of DS associated with CHD may influence the decision to continue the pregnancy. Information from fetal imaging is crucial during consultation, where the option of terminating the pregnancy is often discussed.

In newborns, the postnatal diagnosis of DS is confirmed with genetic tests such as FISH, followed by complete karyotyping within 1 to 2 weeks. Infants with SD should be examined for CHD and ideally undergo echocardiography [25]. When echocardiography is not available, physical examination, ECG, and chest X-ray can be used to improve the initial assessment.

Surgical treatment of CHD in DS patients depends on the type of heart defect, severity of the condition, and comorbidities. The timing and type of surgery vary; for defects such as ASD, VSD, and PDA, the size of the defect and associated conditions are considered. In neonates with TF, surgery is usually performed within 4–6 months, but for more complex physiologies, multiple surgeries may be necessary throughout life, increasing the degree of perioperative risk.

### 4.2. Trisomy 18 or Edwards Syndrome

Edwards syndrome (ES), also known as trisomy 18 (Table 3), is a numerical chromosomal abnormality characterized by three chromosomes 18; it is the most common chromosome abnormality after DS, with a high mortality rate, and is secondary to associated lethal malformations, and only 4% of patients survive the first year of life [26,27].

The prevalence of ED varies globally, with the prevalence of live births ranging from 1 in 3600 to 1 in 10,000 and is greater in females than in males (3:2) [28]. However, the prevalence of live infants with trisomy 18 has increased in recent decades, primarily due to the increase in pregnancies in women with advanced maternal age >35 years [29]. The main causes of mortality are cardiomyopathy, cardiac failure, and respiratory failure [30]. Survival statistics show that 42% of infants survive the first week, 29% survive the first month, 12% survive three months, and only 8% reach six months of life [31].

ED presents a wide range of multisystem clinical manifestations, with more than 125 related anomalies reported, but none of these clinical features are pathognomonic [28].

Cardiac defects are found in 90% of patients with VSD or ASD, PDA, TF, overriding of the aorta, aorta coarctation (AC), hypoplastic left heart syndrome (HLHS), and especially polyvalvular heart disease of the aortic valve and pulmonary valve [28], which is considered by some authors to be a characteristic finding [32].

The evaluation and diagnosis of ED begins during the prenatal period, with the screening of maternal serum, which can detect low levels of alpha-fetoprotein, human chorionic gonadotropin, and unconjugated estriol [28]. These serum and genetic markers are more effective when combined with classical ultrasound findings, such as increased nuchal translucency. If prenatal screening indicates a high risk for fetal aneuploidy, amniocentesis or chorionic villus sampling is recommended [33].

After birth, the evaluation of ED is guided by phenotypic variation and clinical presentation, especially imaging studies such as ultrasounds and echocardiograms, which may be helpful depending on the specific circumstances [34]. Diagnosis is clinical but karyotyping and microarray testing can confirm trisomy and provide details of mosaicism.

Prenatal counseling for ES is complex and involves ethical issues related to high mortality and serious disability. Parents must make difficult choices about resuscitation, life support, and treatments, requiring accurate information on survival and comorbidities. Additionally, they should also be aware of the 1% recurrence risk, which can reach 20% for partial trisomy [28].

### 4.3. Trisomy 13 or Patau Syndrome

Patau syndrome (PS), also known as trisomy 13 (Table 4), is a numerical chromosomal abnormality characterized by three copies of chromosomes 13; it is the third most frequent aneuploidy, with a prevalence of 1 in 18,000 [35], an incidence of 1 in 10,000–20,000 live births, a prenatal mortality of over 95% of pregnancies, and a postnatal survival rate ranging from 6 to 12% beyond the first year of life [36,37].

Advanced maternal age is a significant risk factor due to the increased frequency of chromosomal nondisjunction events [38]. However, it is important to note that approximately 20% of PS cases are attributable to an unbalanced translocation, whereas mosaicism is a rarer cause [39].

The spectrum of heart disease associated with PS includes VSD and ASD, TF, atrioventricular septal defects (AVSDs), and a double-outlet right ventricle (DORV), and, even if untreated, these heart defects are rarely fatal in childhood [38].

The initial assessment of PS begins with the measurement of fetal nuchal translucency between 11 and 14 weeks of gestation, which is usually greater than 3.5 mm [40]. First-trimester screening also includes measurements of B-hCG and PAPP-A, both of which are decreased [40]. Non-invasive prenatal testing (NIPT) can differentiate trisomies 13, 18, and 21, but only amniocentesis and chorionic villus sampling are necessary for the fetal diagnosis [41]. However, a certain diagnosis is only obtained with postnatal karyotyping and the FISH technique [42].

Recent studies indicate that the high mortality associated with trisomies 13 and 18 leads to the termination of 55% of pregnancies with confirmed diagnoses [43]. In fact, both prenatal and postnatal genetic counseling are important for management and for increasing the risk of recurrence in future pregnancies [37,38].

### 4.4. Monosomy X Syndrome or Turner Syndrome

Turner syndrome (TS), also known as monosomy X (Table 5), is a numerical chromosomal abnormality characterized by the deletion or nonfunctioning of one X chromosome (45,XO) in females, and, in 50% of cases, it has a mosaic chromosome component (45. X with mosaicism) [44].

TS is the second genetic cause of congenital heart disease in females, with a prevalence of approximately 1 in 2500 female live births [11], although the true prevalence is unknown since patients with a mild phenotype may remain undiagnosed or are diagnosed in late adulthood [45].

The most commonly associated heart defects include bicuspid aortic valve (BAV) and AC, which, together with systemic hypertension, can lead to aortic dilation and dissection, increasing the risk of SCD [11,46].

BAV is a congenital anomaly that occurs in up to 30% of patients with TS, often without obvious symptoms [47]. It is important to identify BAVs in asymptomatic patients because they are at an increased risk for complications such as infective endocarditis, significant valvular stenosis, valvular regurgitation, and aortic aneurysm [47]. For this reason, regular medical evaluation is needed, with echocardiography as the primary screening test, and CMR may be necessary for detailed visualization of the aortic valve. In severe or complicated cases, preventive surgery may be considered to reduce the risk of aortic dissection or rupture.

AC affects approximately 12% of women with TS [47] and is usually diagnosed clinically in childhood with hypertension and brachial‒femoral retardation. For an accurate diagnosis, a cardiac magnetic resonance (CMR) angiography or computed tomography (CT) scan is recommended [48]. The narrowing of the aorta causes pressure gradients that can lead to serious complications, such as hypertension, congestive heart failure, dissection, and aortic rupture. Furthermore, even after surgery, there is an elevated risk of hypertension, CHD, cerebrovascular disease, aortic dissection, and restenosis persists [49].

An elongated transverse arch (ETA) is an abnormality detected by CMR and is characterized by an increase in the distance between the origin of the left common carotid artery and the left subclavian artery, with the flattening of the arch and kinking [47]. An ETA is embryologically like coarctation and may predispose patients to complications such as aortic dilation and dissection but differs from true coarctation in that it does not present luminal narrowing, pressure gradients, or collateral circulation [50]. An ETA is often associated with BAV, AC, and the dilatation of the aortic sinus [47].

Aortic dilation affects 23% of women with TS [51] and, less commonly, approximately 5% of women with TS alone. This necessitates periodic monitoring for all women with TS [52]. Echocardiographic evaluation is recommended annually for those with an enlarged aortic root and every 2 to 3 years for those with normal aortic root dimensions. An aortic surface area index (ASI) ≥ 2 cm/m^2^ indicates the need for careful monitoring, whereas an ASI ≥2.5 cm/m² requires surgical intervention to prevent aortic dissection [53].

Aortic dissection is a significant concern in TS, with an incidence of 40 cases per 10,000 patients and often resulting in fatality [47]. It occurs at an early age, with a median age of 35 years, and has the highest incidence rates between the ages of 20 and 39 [54].

Furthermore, venous anomalies, such as partial anomalous pulmonary venous connections and persistent left superior vena cava, are also described in TS patients. Cardiac conduction defects are common in TS, often due to left ventricular hypertrophy, myocardial ischemia, previous infarcts, and congenital cardiac malformations. Other associated lesions may include ventricular and atrial septal defects, hypoplastic left heart syndrome, a single ventricle, mitral valve abnormalities, coronary artery abnormalities, and an aberrant right subclavian artery [47].

TS can be diagnosed prenatally via chorionic villus sampling or amniocentesis. The presence of the syndrome should be suspected if signs such as hydrops fetalis, cystic hygroma, or heart defects are detected during prenatal ultrasound. To confirm the diagnosis, a karyotype test should be performed during the pregnancy. In some cases of mosaicism, the karyotype may be normal, so if the clinical suspicion is strong, a fluorescence in situ hybridization (FISH) study can be performed as a further diagnostic investigation.

During adolescence, patients with TS may experience a delayed onset of puberty or amenorrhea with elevated follicle-stimulating hormone (FSH) levels.

When patients are diagnosed with TS, an initial screening for associated pathologies and periodic screening are necessary.

## 5. Structural Chromosomal Abnormalities

### 5.1. 22.q11.2 Deletion or DiGeorge Syndrome

DiGeorge syndrome (DGS) (Table 6) is a structural chromosomal abnormality that results predominantly from the microdeletion of chromosome 22, more specifically on the long arm (q) at the 11.2 locus (22q11.2) and causes the failure to properly develop the pharyngeal pouches, which are responsible for the embryological development of several organs, including the aortic arch and cardiac outflow tract [55].

In most cases, DGS is a de novo mutation without any genetic abnormalities in the proband’s parents [56], and microdeletion is responsible for 90% of cases [55].

DGS affects approximately 0.1% of fetuses [57], with an estimated rate of 1 in 4000–6000 live births [56].

Clinically, DGS is characterized by a wide range of phenotypes, the most common of which include cardiac abnormalities, hypocalcemia, and hypoplastic thymus.

The prevalence of CHD in DGS varies between 48.5% and 79% depending on the population and diagnostic context. The most recent studies reported a lower prevalence than previous studies, which reported 75–80% prevalence rates. The prevalence of CHD is affected by the age of diagnosis and is higher in fetuses and newborns than in older children [58].

The most common cardiac defects in DGS are conotruncal defects, such as tetralogy of Fallot and truncus arteriosus, ventricular septal defects, and atrial septal defects [59,60]. Aortic arch anomalies, both isolated and combined with intracardiac defects, such as those of the cervical aortic arch, double aortic arch, and straight aortic arch, as well as variations in the origin of the subclavian arteries, are frequent [61,62].

Rare cases of other CHDs, such as hypoplastic left heart syndrome, heterotaxy, and valvar pulmonary stenosis, have also been reported. Additionally, a subgroup of patients presents with aortic root dilation, which may be progressive, although the clinical significance of this condition remains uncertain. Patients with tetralogy of Fallot and DGS are at greater risk of aortic root dilatation than those without deletion [58].

Microdeletions causing DGS are detected via techniques such as fluorescent in situ hybridization (FISH), multiplex ligation-dependent probe amplification (MLPA), single nucleotide polymorphism (SNP) arrays, comparative genomic hybridization (CGH) microarrays, and quantitative polymerase chain reaction (qPCR) [55].

Patients diagnosed with or suspected of DGS should undergo thorough evaluations, especially if they have life-threatening cardiac deficiencies. Cardiac anomalies, if not identified during fetal ultrasound, can manifest soon after birth as life-threatening cyanotic heart disease. In such cases, an urgent evaluation for pediatric cardiothoracic surgery may be necessary, which varies depending on the heart defect.

### 5.2. q11.23 Deletion or Williams Syndrome

Williams syndrome (WS) (Table 7) is a structural chromosomal abnormality caused by a deletion in the 7q11.23 band involving the elastin gene (ELN) and is a rare genetic disease associated with congenital heart disease [63].

WS has an estimated frequency ranging from 1 in 7500 to 1 in 75,000 children [64] and affects both sexes equally [65].

Cardiovascular abnormalities are present in 80% of SW patients and are the main cause of morbidity and mortality [66].

In WS, cardiovascular abnormalities are frequent and can manifest in a variety of ways. The first clinical sign is often a heart murmur, leading to further investigations. In children with WS, supravalvular aortic stenosis (SVAS), characterized by a narrowing of the aorta above the aortic valve, and pulmonary artery stenosis, which can affect both the main and branch arteries, are common [64].

In adults with WS, pulmonary vascular disease tends to be less noticeable than it is in children. However, other cardiovascular abnormalities can be present at any age. These include stenoses in other blood vessels, cardiac septal defects, hypertension, vascular stiffness, and ECG abnormalities. While these conditions are generally not the primary reason for referral, they can still be detected during diagnosis and require monitoring and appropriate management [64].

The diagnosis of WS is based on the identification of a heterozygous 1.5–1.8 Mb deletion on chromosome 7q11.23, and this recurrent deletion is located at position chr7:73,330,452–74,728,172 of the reference genome [67]. Although genes such as ELN are included in the deletion, no single gene has been identified as causal for WS [67].

The genomic analysis techniques used to determine sequence copy number include a chromosomal microarray (CMA) and FISH.

A CMA, which uses oligonucleotide arrays or SNP genotyping, can detect recurrent deletions, and the size of the deletion depends on the type of microarray and the density of the probes.

FISH targeting the 7q11.23 region is robust for diagnosis when a CMA is not available and can be used to test at-risk relatives. However, it is not suitable for relatives of individuals suspected to have WS without a deletion confirmed by FISH or a CMA.

At birth, infants with Williams syndrome often require cardiac care owing to supravalvular aortic stenosis, which may require open-heart surgery performed by a cardiothoracic surgeon. After surgery, close monitoring by a cardiologist is essential, given the risk of hypertension and arterial disease. These conditions can also lead to complications such as pulmonary artery stenosis, mitral valve insufficiency, and renal artery stenosis, which require ongoing clinical management.

Genetic counseling is crucial in WS, an autosomal dominant disease. Most cases of WS are caused by a de novo 1.5–1.8 Mb deletion in the 7q11.23 region, although it is rare for an affected parent to pass the condition on to their children. Recommendations for parents of a child with WS include obtaining a detailed medical history to screen for any signs or symptoms of the syndrome. If there are no clinical signs of WS in the parents, testing for the 7q11.23 deletion is usually not recommended [67].

Each child of an individual with WS has a 50% chance of inheriting the 7q11.23 deletion and developing the syndrome. Once the 7q11.23 deletion causing WS has been identified in a family member, prenatal and preimplantation genetic testing may be considered to assess the risk of passing the condition on to future children [67].

## 6. Sudden Cardiac Death in Children

CHD is among the most prevalent congenital anomalies in newborns. While exceedingly rare, these abnormalities can result in SCD during childhood. SCD is a catastrophic event characterized by natural cardiovascular-related death, marked by a sudden loss of consciousness within an hour of the onset of cardiac symptoms [7,68].

Despite progress in prevention strategies, cardiac death remains the leading cause of death worldwide [68]. SCD represents a significant international public health problem, constituting a considerable challenge for modern medicine, especially when it affects young people [7,68].

SCD in young patients is a tragic event that profoundly affects families and communities, accounting for up to 25% of deaths among those affected by CHD [69]. Accurately describing SCD in CHD is complicated by geographic variations in reporting, errors in death certificates, lack of autopsies, and inaccuracies in diagnosis codes.

The incidence of SCD among individuals up to 35 years of age requires particular attention and varies depending on the age considered [7].

SCD in CHD patients is a rare event in the general population, with an annual incidence rate of 0.07–0.40 per 100,000 person-years [5,6]. However, in CHD patient cohorts, SCD affects 0.28–2.7% of patients annually, which is significantly higher than that reported in the general population [5,8]. The risk of SCD is especially high in patients with tetralogy of Fallot (approximately 0.9–1.5% per year) [4], transposition of the great arteries, cyanotic heart disease, Ebstein anomaly, and Fontan circulation. The incidence of SCD is greater in adults than in children, with peaks occurring between the ages of 30 and 40 years in patients with complex CHD and after the age of 50 years in those with simple forms of CHD [8]. It is suspected that 5–10% of SCD cases in these patients are attributable to congenital heart disease and primary genetic electrical disorders [11].

Notably, the cause of death remains unknown in 33% of cases of sudden death, despite autopsy examination, and in such cases, a postmortem genetic study called molecular autopsy is necessary [7,70].

When a patient with CHD dies unexpectedly, it is crucial that cardiologists and pathologists collaborate closely to determine the exact cause of death. Autopsies revealed that in a series of postoperative cases with CHD, 8.5% had abnormalities that, if known before death, could have significantly affected clinical treatment [4].

It is essential to emphasize the need to perform more extensive autopsies in the setting of SCD, especially because a significant portion of SCD events in CHD patients occur before the diagnosis of the congenital disease [71].

This highlights the importance of autopsies not only to support affected families and learn from the circumstances of death but also to improve future medical care through audits and education.

Our review of the literature revealed that numerical chromosomal anomalies, such as DS, ES, PS, and TS, can be associated with CHD defects and therefore an increased risk of SCD. These conditions pose serious medical challenges, requiring timely diagnosis and careful management to improve patients’ quality of life.

DS is the most common of these anomalies, with 16 cases per 10,000 live births. It is associated with various CHDs, which are found in 40–50% of newborns with DS, a much higher incidence than the 1% reported in the general population. The most frequent defects include AVSD and VSD. Other common defects include TF, PDA, and ASD. These heart defects increase the risk of coronary events and SCD.

ES is the second most common numerical chromosomal anomaly and has a very severe prognosis, with only 4% of newborns surviving the first year of life. Approximately 90% of patients with ES have cardiac defects, including VSD, ASD, PDA, and TF, as well as rarer conditions such as AC and HLHS. Cardiomyopathy, heart failure, and respiratory failure are major causes of mortality, contributing significantly to the risk of SCD.

PS is characterized by very high prenatal mortality and very low postnatal survival, with only 6–12% of newborns surviving beyond the first year of life. Associated heart defects include VSD, ASD, TF, and right ventricular double outlet. Although rarely fatal in childhood without treatment, these defects can lead to severe cardiac complications and a high risk of SCD.

TS affects approximately 1 in 2500 newborn girls. Patients with ST may have cardiac defects such as BAV and AC. These conditions can lead to serious complications, such as aortic dilation and dissection, increasing the risk of SCD. Aortic dilatation is common in women with TS and requires regular monitoring to prevent fatal aortic dissections.

Prenatal diagnosis of numerical chromosomal abnormalities is essential for the management of associated CHD. Prenatal screening includes ultrasound, maternal blood tests, and non-invasive prenatal testing. The definitive diagnosis is confirmed by amniocentesis or chorionic villus sampling. After birth, the postnatal diagnosis is confirmed via genetic tests such as karyotyping and FISH.

The management of numerical chromosomal abnormalities requires a multidisciplinary approach. Cardiac surgery is often necessary to correct defects such as ASD, VSD, and PDA. For more complex conditions such as TF, multiple interventions may be necessary over the patient’s lifetime. In TS, the close monitoring of the aorta and heart valves is essential to prevent serious complications.

Numerical chromosomal anomalies are a significant cause of CHD and SCD. Early diagnosis and accurate management are essential to improve quality of life and reduce mortality in affected patients. Understanding the clinical implications and implementing appropriate monitoring and treatment strategies are critical to addressing the challenges posed by these genetic conditions.

Our literature review revealed that structural chromosomal anomalies, such as DGS and WS, are associated with a high risk of CHD and SCD.

DGS involves a failure to properly develop the pharyngeal pouches, with significant implications for the formation of the aortic arches and cardiac outlet tract. This malformation can lead to a variety of congenital heart defects, including conotruncal anomalies such as tetralogy of Fallot and truncus arteriosus, as well as ventricular and atrial septal defects. The prevalence of CHD in DGS ranges from 48.5% to 79%, with a higher incidence in fetuses and newborns than in older children. Additionally, aortic arch abnormalities and aortic root dilation have been observed, although the latter is not always well understood clinically.

The identification of the microdeletion responsible for DGS can be carried out through advanced genomic analysis techniques, such as FISH and array CGH, which are essential for early diagnosis and timely intervention. The clinical management of DGS, particularly in patients with severe cardiac abnormalities, requires thorough evaluation and urgent cardiac surgeries to prevent fatal complications.

WS is associated with a significant incidence of cardiovascular abnormalities, with a cardiac involvement rate of up to 80%. The most common manifestations include supravalvular aortic stenosis and pulmonary artery stenosis, which may require complex surgical interventions. SW also has other cardiovascular abnormalities that can manifest in adulthood, such as hypertension and vascular stiffness.

Long-term monitoring of patients with WS is critical to managing potential cardiovascular complications and preventing adverse events. The diagnosis is based on the detection of the deletion at 7q11.23 via techniques such as CMA and FISH. Genetic counseling is crucial, as the syndrome can be inherited, and parents of affected children must be informed about the risks and prenatal and preimplantation screening options.

Both syndromes demonstrate the importance of accurate cardiac and genetic evaluation in the presence of structural CAs. Timely and targeted management of associated CHD can significantly improve the quality of life and reduce the risk of sudden cardiac death in affected patients. In-depth knowledge of clinical manifestations and diagnostic and therapeutic options is essential for optimal care and a favorable prognosis.

CAs contribute to the development of CHD through genetic and epigenetic mechanisms that alter normal cardiac morphogenesis. Recent studies highlight that copy number variations (CNVs) and mutations of regulatory genes are among the main molecular determinants. Genes such as NKX2.5, GATA4, and TBX5, which encode key transcription factors, have been identified as responsible for complex congenital defects, including septal anomalies and aortic arch malformations; the 22q11.2 locus, frequently associated with DiGeorge syndrome, involves critical genes such as TBX1, which is essential for the development of the cardiac outflow tract [72,73,74].

Advanced techniques such as whole exome sequencing (WES) and the analysis of CNVs via a microarray or CNV-seq techniques are transforming prenatal diagnosis, allowing the identification of genetic anomalies not detectable with traditional karyotyping. These techniques have significantly improved the rate of genetic detection in CHD, especially in cases with associated extracardiac malformations, and allow for personalized clinical management, including reproductive counseling and early intervention [73,74].

Understanding these anomalies not only supports targeted treatment but offers opportunities for emerging therapies such as gene editing, which aims to correct specific mutations during prenatal development.

In recent years, advances in genetic engineering and gene therapy revolutionized the treatment of birth defects, including those associated with CAs that affect the heart.

Research is focusing on identifying specific genetic variants linked to CHD. The use of advanced technologies such as genome and exome sequencing has identified many responsible gene variants, including genes involved in the regulation of chromatin, transcription factors, and cellular signaling [75].

CRISPR-associated protein 9 (CRISPR-Cas9) has been used successfully to correct genetic mutations in several diseases and is also being studied to treat CHD [76,77]. This technique allows cutting the DNA at the precise point where the pathogenic mutation is present, allowing the cell to repair the genetic defect and offering prospects for personalized medicine while highlighting the importance of confirming genomic editing and preventing unwanted effects.

Gene therapy using viral vectors is already used in approved treatments for rare genetic diseases [78], but researchers are developing nonviral vectors to reduce the costs and risks associated with immune responses [79]. An emerging field is gene therapies administered directly to the fetus during pregnancy, with the aim of correcting genetic defects before birth [80,81].

Despite progress, the large-scale clinical application of these technologies still presents challenges, such as the safety of treatments, high cost, and the need for larger clinical trials to evaluate efficacy and long-term risks.

## 7. Prevention Strategies in Public Health

SCD in children with CHD represents a major international public health problem (Figure 1), both because of its devastating and unpredictable nature and the significant burden it imposes on health systems and communities.

In fact, despite the progress of modern medicine, SCD remains a complex and difficult event to prevent, and its impact goes far beyond individual loss because it not only destroys the life of a family but affects entire communities and highlights the shortcomings in the prevention and treatment of congenital heart defects.

Furthermore, geographic disparities in case reporting and failure to require autopsies [82] contribute to an incomplete understanding of SCD, limiting the effectiveness of preventive measures worldwide.

SCD requires a multidisciplinary approach, including cardiologists, geneticists, surgeons, pathologists, and public health specialists, various levels of the health system, and different prevention strategies to be implemented in public health.

Advanced prenatal diagnosis with prenatal screening, such as fetal echocardiography and genetic testing, can be used to identify cardiac defects and chromosomal abnormalities that predispose newborns to a high risk of SCD.

Postnatal monitoring, where children diagnosed with CHD must undergo continuous and rigorous monitoring, can be implemented with regular cardiology visits to detect early arrhythmias or other cardiac abnormalities.

Timely surgical interventions and personalized treatments, for children with complex CHD, are essential to correct the structural abnormalities of the heart and reduce the risk of SCD, which should be supported by targeted pharmacological therapies and, in the most severe cases, by the implantation of devices such as implantable cardiac defibrillators to prevent potentially fatal malignant arrhythmias.

Education and awareness among families and caregivers on the warning signs that may precede a sudden cardiac event, such as palpitations, fainting, or difficulty breathing, are essential.

Molecular autopsy, in cases of unexplained SCD, can reveal predisposing genetic abnormalities that were not diagnosed during life, not only to clarify the causes of death but also to provide valuable data to improve prevention and the treatment of family members.

## 8. Discussion of Results and Implications for Public Health

This state-of-the-art review highlights how chromosomal abnormalities are closely linked to CHD and the increased risk of SCD in children. Among these anomalies, both the numerical ones such as trisomy 21, 18, and 13, Turner syndrome, and the structural ones such as DiGeorge syndrome and Williams syndrome stand out.

We critically analyze these findings from a clinical and public health perspective, with a particular focus on future developments and preventive strategies.

Regarding early diagnosis and therapeutic impact, advanced prenatal screening techniques, such as fetal echocardiography and non-invasive genetic testing, are essential to promptly identify chromosomal abnormalities associated with CHD. These tools allow for targeted clinical management, improving the possibility of early and personalized intervention. However, the accessibility of these technologies is often limited by socioeconomic factors, creating inequalities in care.

As for survival and the management of CHD, although advances in cardiovascular surgery have increased survival in patients with CHD, the clinical outcome is influenced by the presence of extracardiac malformations and genetic etiology. For example, while newborns with trisomy 21 can benefit from specific surgical interventions, those with trisomy 18 or 13 have a more severe prognosis, limited by the complexity of the malformations and high early mortality.

As for public health involvement, the prevention and management of CHD require a multidisciplinary approach. It is essential to promote awareness campaigns to improve awareness of the risks associated with CHD and SCD. Molecular post-mortem genetic study, called molecular autopsy, can allow us to understand the genetic mechanisms underlying SCD, contributing to prevention in the future.

The use of techniques such as genome sequencing could broaden the understanding of chromosomal anomalies and their clinical manifestations, allowing increasingly personalized and early interventions.

Large-scale non-invasive technologies may lead to the implementation of non-invasive screening that could significantly reduce diagnostic delays, particularly in resource-limited settings.

Continuous monitoring consisting of wearable cardiac monitoring devices could represent a valid method to support the timely detection of potentially lethal abnormalities in patients with CHD.

Improving the early diagnosis and management of CHD represents a significant public health benefit. In addition to reducing infant mortality, targeted preventive strategies could reduce the burden on health facilities while promoting greater equity in access to health services. Furthermore, international collaboration could facilitate the creation of global registries on CHD and SCD, allowing epidemiological trends to be monitored and the effectiveness of adopted interventions to be evaluated.

## 9. Conclusions

SCD in children with CHD is not just a personal tragedy, but a public health crisis that requires collective and coordinated action. Understanding chromosomal abnormalities and their careful management are essential to reduce the risk of sudden cardiac death and improve the quality of life of affected patients.

Collaboration among a multidisciplinary team, such as cardiologists, geneticists, surgeons, pathologists, and public health specialists, is essential to ensure a precise diagnosis and adequate treatment.

## Figures and Tables

**Figure 1 medicina-60-01976-f001:**
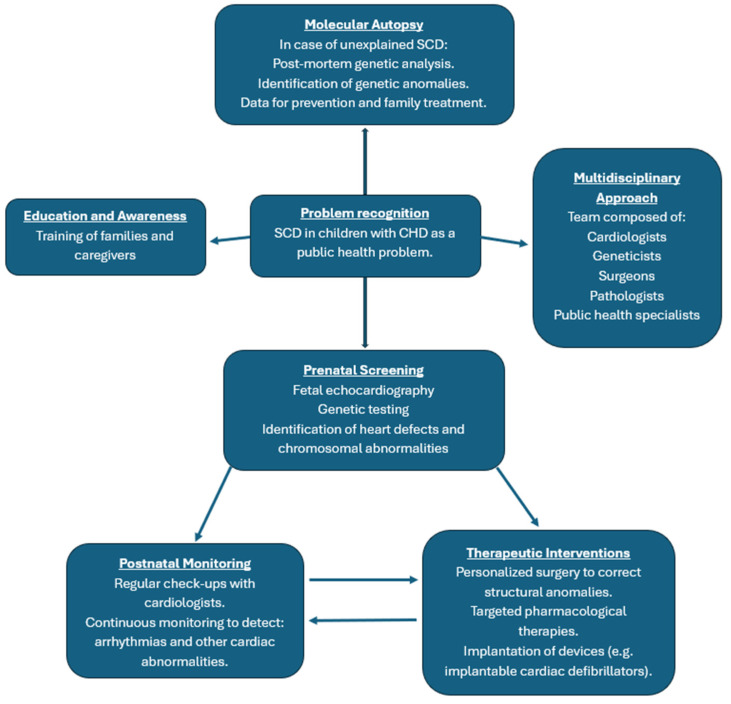
Flowchart on preventive strategies in public health.

**Table 1 medicina-60-01976-t001:** Chromosomal abnormalities associated with cardiovascular diseases.

Chromosomal Abnormalities	Description	Syndrome	Cardiovascular Diseases Associated (More Frequent)
Numerical			
Aneuploidies	Abnormal number of chromosomes		
Trisomy	An extra chromosome (3 copies instead of 2)	Trisomy 21 or Down Syndrome	atrioventricular septal defects and ventricular septal defects
		Trisomy 18 or Edwards Syndrome	polyvalvular heart disease
		Trisomy 13 or Patau’s Syndrome	ventricular and atrial septal defects
Monosomy	Missing a chromosome	Monosomy X or Turner Syndrome	bicuspid aortic valve and coarctation of the aorta
Structural			
Microdeletion	Small deletions that can be difficult to detect with standard techniques	7q11.23 deletion or Williams Syndrome	supravalvular aortic stenosis and/or pulmonary artery stenosis
Deletions	Loss of a portion of a chromosome	22q11.2 deletion or DiGeorge Syndrome	conotruncal defects

**Table 2 medicina-60-01976-t002:** Congenital heart disease in Down syndrome.

Characteristic	Description
Definition	numeric chromosomal abnormality characterized by three chromosomes 21
Epidemiology	incidence 16 cases every 10,000 live births most frequent chromosomal anomaly
CHD association	SD is the chromosomal abnormality most frequently associated with CHD: atrioventricular septal defects ventricular septal defects atrial septal defects patent ductus arteriosus tetralogy of Fallot
CHD probability	patients with DS are 40–50 times more likely to develop CHD compared to the general population
CHD more frequent	AVSD and VSD account for 76% most frequent AVSD isolated VSD 4–17% isolated ToF 13% combined AVSD and ToF 9%
Further risk factors for CHD	maternal smoking, obesity, and folate deficiency during pregnancy
CHD incidence on ultrasonography	29–56% in cases of DS confirmed by karyotyping
Examinations for CHD in newborns	echocardiography recommended
Surgical management of CHD	timing and type of intervention vary depending on the type and severity: defects such as ASD, VSD, and PDA, the size of the defect and the associated conditions are considered infants with TF often operated on within 4–6 months more complex physiologies, multiple interventions throughout life

**Table 3 medicina-60-01976-t003:** Congenital heart disease in Edwards syndrome.

Characteristic	Description
Definition	numeric chromosomal abnormality characterized by the presence of three chromosomes 18
Epidemiology	more common after DS prevalence 1 in 3600 to 1 in 10,000 live births higher in females than in males (3:2 ratio)
Survival	42% first week 29% first month 12% three months 8% six months 4% first year of life
Causes of death	cardiomyopathy, cardiac failure, respiratory failure
CHD association	heart defects present in 90% of patients: ventricular or atrial septal defect patent ductus arteriosus tetralogy of Fallot left heart hypoplasia multivalvular disease coarctation of the aorta.
Genetic counseling	recurrence risk: 1% for full trisomy, up to 20% for partial trisomy

**Table 4 medicina-60-01976-t004:** Congenital heart disease in Patau syndrome.

Characteristic	Description
Definition	numeric chromosomal abnormality characterized by the presence of three chromosomes 13
Epidemiology	third most frequent aneuploidy prevalence 1 in 18,000 incidence 1 in 10,000–20,000 live births prenatal mortality more than 95% of pregnancies
Survival	6–12% beyond the first year of life
CHD association	ventricular and atrial septal defects tetralogy of Fallot atrioventricular septal defects double outflow of the right ventricle
Genetic counseling	recurrence risk in future pregnancies

**Table 5 medicina-60-01976-t005:** Congenital heart disease in Turner syndrome.

Characteristic	Description
Definition	numerical chromosomal anomaly characterized by the deletion or nonfunctioning of an X chromosome
Epidemiology	prevalence of approximately 1 in 2500 live births
CHD association	bicuspid aortic valve coarctation of the aorta elongated Transverse Arch ventricular/atrial septal defects hypoplastic left heart syndrome mitral valve anomalies
CHD more frequent	bicuspid aortic valve: up to 30% coarctation of the aorta approximately 12%
Venous Anomalies	anomalous partial pulmonary venous connection persistent left superior vena cava
Complications	aortic dilatation aortic dissection conduction defects

**Table 6 medicina-60-01976-t006:** Congenital heart disease in DiGeorge syndrome.

Characteristic	Description
Definition	microdeletion of chromosome 22 on the long arm (q) at locus 11.2
Epidemiology	approximately 0.1% of fetuses estimated rate of 1 in 4000 to 6000 live births CHD prevalence: varies between 48.5% and 79%, higher in fetuses and newborns than in older children
CHD association	conotruncal defects such as tetralogy of Fallot and truncus arteriosus ventricular and atrial septum defects cervical aortic arch double aortic arch straight aortic arch variations in the origin of the subclavian arteries hypoplastic left heart syndrome heterotaxy valvular pulmonary stenosis
Recommended assessments	extensive testing for life-threatening cardiac abnormalities that may require urgent pediatric cardiothoracic surgery

**Table 7 medicina-60-01976-t007:** Congenital heart disease in Williams syndrome.

Characteristic	Description
Definition	deletion in band 7q11.23 of chromosome 7, involving the elastin (ELN) gene
Epidemiology	frequency estimated from 1 in 7500 to 1 in 75,000 children cardiovascular anomalies present in 80% of patients are the main cause of morbidity and mortality
CHD association	supravalvular aortic stenosis (SVAS) pulmonary artery stenosis cardiac septal defects
Other cardiac anomalies	hypertension vascular rigidity ECG anomalies
Genetic counseling	autosomal dominant disease, recommended for assessing the risk of transmission to children

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
