# Peer review of "Cardiovascular Diseases in Public Health: Chromosomal Abnormalities in Congenital Heart Disease Causing Sudden Cardiac Death in Children"

_medicina, 2024, doi:10.3390/medicina60121976_

Round 1

Reviewer 1 Report

Comments and Suggestions for Authors

The authors reviewed the structural and functional CAs responsible for congenital heart disease (CHD) that increase the risk of SCD and analyze the prevention strategies to be implemented to reduce SCD.

-              The study is novel and well fits in the context of the current literature.

-              The introduction is too short lacks appropriate background and should be expanded, better focusing on literature content that justify this review article.Explain why chromosomal abnormalities are important to be targeted in congenital heart disease causing Sudden Cardiac Death in Children. Which issues for public health?

-               Although it is a general review, it is important to add a methods paragraph dealing with inclusion and exclusion criteria, databases used, etc, It is important to understand the criteria for articles inclusions.

-              Before conclusions, I would add a discussion section that critically analyses the findings of the study, also linking them to new perspectives and the possible advantages for public health.

Author Response

Dear Reviewer1,

Thank you for your comments that improve the article and as indicated:

Comments 1: The introduction is too short lacks appropriate background and should be expanded, better focusing on literature content that justify this review article.Explain why chromosomal abnormalities are important to be targeted in congenital heart disease causing Sudden Cardiac Death in Children. Which issues for public health?

Response 1: We added paragraphs in the introduction to explain the importance of studying CAs to prevent CHD and SCD and the implications for public health. We also better explained the purpose of our review. (highlighted in yellow)

Comments 2: Although it is a general review, it is important to add a methods paragraph dealing with inclusion and exclusion criteria, databases used, etc, It is important to understand the criteria for articles inclusions.

Response 2: We added paragraph 3 “Materials and methods”: lines 96-103 and highlighted in yellow.

Comments 3: Before conclusions, I would add a discussion section that critically analyses the findings of the study, also linking them to new perspectives and the possible advantages for public health.

Response 3: We added paragraph 8. “Discussion of Results and Implications for Public Health”: lines 546-588 and highlighted in yellow.

Kind regards.

Reviewer 2 Report

Comments and Suggestions for Authors

1. Summary of the Manuscript

This manuscript reviews the relationship between chromosomal abnormalities (CAs) and congenital heart disease (CHD), which leads to an increased risk of sudden cardiac death (SCD) in children. The authors provide a comprehensive overview of the various types of CAs, such as trisomies and monosomies, and their association with different types of CHD. The paper emphasizes the public health implications of these conditions and discusses diagnostic strategies and preventive measures to mitigate the risks associated with these genetic conditions.

2. Major Strengths

  1. Relevance: The topic is highly relevant to current public health concerns and the management of congenital heart diseases.
  2. Comprehensiveness: The review covers a broad range of chromosomal abnormalities and links these with specific congenital heart defects, providing a thorough overview.
  3. Clinical Implications: The discussion on the implications for diagnosis and management of these conditions is well-articulated and informative for clinicians and researchers alike.

3. Major Concerns

  1. Depth of Review: While the manuscript covers a broad range of topics, it lacks depth in certain areas, particularly in discussing the molecular mechanisms underlying these chromosomal abnormalities and their direct impact on cardiac development.
  2. Currentness of Literature: Some references are quite outdated. Inclusion of more recent studies could provide a better understanding of the advances in genetic testing and management strategies.
  3. Statistical Analysis: The paper lacks a critical analysis of the studies reviewed, such as the strength of the associations reported and the statistical significance of findings across studies.
  4. Figures and Tables: While the tables are informative, additional graphical representations (figures or flowcharts) of the genetic pathways or a summary of diagnostic strategies could enhance understanding and readability.

4. Specific Recommendations

  • Expand Discussion: Include a section on recent genetic engineering or therapy advancements that could potentially address these congenital defects.
  • Update Literature: Ensure that all cited studies are current and include literature up to the year of publication.
  • Enhance Statistical Commentary: Provide a more detailed analysis of the data presented in the review, possibly including a meta-analysis if applicable.
  • Improve Visual Elements: Add figures or diagrams to summarize the complex information, especially the pathways affected by different chromosomal abnormalities.

5. Ethical, Confidentiality, and Conflict of Interest

No ethical issues or conflicts of interest noted. The reviewer maintains confidentiality by MDPI policies.

6. Conclusion

The manuscript is promising but requires significant revisions to deepen the content and update the literature to meet the journal's standards. The subject is of great importance, and with the suggested improvements, the paper could provide valuable insights into the prevention and management of CHD associated with chromosomal abnormalities.

Author Response

Dear Reviewer2,

Thanks for the comments that improve the article and as indicated:

Comments 1: Depth of Review: While the manuscript covers a broad range of topics, it lacks depth in certain areas, particularly in discussing the molecular mechanisms underlying these chromosomal abnormalities and their direct impact on cardiac development. Expand Discussion: Include a section on recent genetic engineering or therapy advancements that could potentially address these congenital defects.

Response 1: We have added in paragraph 6 “Sudden Cardiac Death in Children” the following sections:

---a section on the mechanisms underlying cardiac morphogenesis: lines 473-489, light blue highlighted, citations 79-81;

---a section on recent advances in genetic engineering or gene therapy: lines 459-479 and green highlighted, citations 82-88.

Comments 2: Currentness of Literature: Some references are quite outdated. Inclusion of more recent studies could provide a better understanding of the advances in genetic testing and management strategies. Update Literature: Ensure that all cited studies are current and include literature up to the year of publication.

Response 2: We have updated older citations with more recent references (highlighted in red in the text and references) and added new citations for the added paragraphs.

Comments 3: Statistical Analysis: The paper lacks a critical analysis of the studies reviewed, such as the strength of the associations reported and the statistical significance of findings across studies. Enhance Statistical Commentary: Provide a more detailed analysis of the data presented in the review, possibly including a meta-analysis if applicable.

Response 3: Meta-analysis/statistical analysis is not applicable since this is a literature review that is not intended for this purpose.

Comments 4: Figures and Tables: While the tables are informative, additional graphical representations (figures or flowcharts) of the genetic pathways or a summary of diagnostic strategies could enhance understanding and readability. Improve Visual Elements: Add figures or diagrams to summarize the complex information, especially the pathways affected by different chromosomal abnormalities.

Response 4: We have added Figure 1 Flowchart on preventive strategies in Public Health.

Best regards.

Round 2

Reviewer 1 Report

Comments and Suggestions for Authors

Amended manuscript is acceptable.

Reviewer 2 Report

Comments and Suggestions for Authors

Every request was promptly attended to.